# Neonatal Lead (Pb) Exposure and DNA Methylation Profiles in Dried Bloodspots

**DOI:** 10.3390/ijerph17186775

**Published:** 2020-09-17

**Authors:** Luke Montrose, Jaclyn M. Goodrich, Masako Morishita, Joseph Kochmanski, Zachary Klaver, Raymond Cavalcante, Julie C. Lumeng, Karen E. Peterson, Dana C. Dolinoy

**Affiliations:** 1Department of Community and Environmental Health, Boise State University, Boise, ID 83725, USA; 2Department of Environmental Health Sciences, University of Michigan, Ann Arbor, MI 48109, USA; gaydojac@umich.edu (J.M.G.); karenep@umich.edu (K.E.P.); ddolinoy@umich.edu (D.C.D.); 3Department of Family Medicine, Michigan State University, East Lansing, MI 48824, USA; tomoko@msu.edu (M.M.); klaverza@msu.edu (Z.K.); 4Department of Translational Neuroscience, Michigan State University, Grand Rapids, MI 49503, USA; jjkoch77@gmail.com; 5Department of Computational Medicine and Bioinformatics, University of Michigan, Ann Arbor, MI 48109, USA; rcavalca@umich.edu; 6Department of Pediatrics, University of Michigan, Ann Arbor, MI 48109, USA; jlumeng@umich.edu; 7Department of Nutritional Sciences, University of Michigan, Ann Arbor, MI 48109, USA

**Keywords:** exposure assessment, epigenomics, neonatal bloodspots, metals, developmental origins of health and disease, biomarkers

## Abstract

Lead (Pb) exposure remains a major concern in the United States (US) and around the world, even following the removal of Pb from gasoline and other products. Environmental Pb exposures from aging infrastructure and housing stock are of particular concern to pregnant women, children, and other vulnerable populations. Exposures during sensitive periods of development are known to influence epigenetic modifications which are thought to be one mechanism of the Developmental Origins of Health and Disease (DOHaD) paradigm. To gain insights into early life Pb exposure-induced health risks, we leveraged neonatal dried bloodspots in a cohort of children from Michigan, US to examine associations between blood Pb levels and concomitant DNA methylation profiles (*n* = 96). DNA methylation analysis was conducted via the Infinium MethylationEPIC array and Pb levels were assessed via high resolution inductively coupled plasma mass spectrometry (HR-ICP-MS). While at-birth Pb exposure levels were relatively low (average 0.78 µg/dL, maximum of 5.27 ug/dL), we identified associations between DNA methylation and Pb at 33 CpG sites, with the majority (82%) exhibiting reduced methylation with increasing Pb exposure (q < 0.2). Biological pathways related to development and neurological function were enriched amongst top differentially methylated genes by *p*-value. In addition to increases/decreases in methylation, we also demonstrate that Pb exposure is related to increased variability in DNA methylation at 16 CpG sites. More work is needed to assess the accuracy and precision of metals assessment using bloodspots, but this study highlights the utility of this unique resource to enhance environmental epigenetics research around the world.

## 1. Introduction

Even with recent reductions in lead (Pb) exposure [1], Pb toxicity and related health effects remain a concern in nearly every country around the world. According to the Housing and Urban Development’s American Healthy Homes Survey, which uses the United States (US) data from 2005 and 2006, almost a quarter of all US homes are estimated to have multiple Pb-based paint hazards [2]. There are also mounting concerns about the risk of exposure to Pb due to the aging infrastructure. The Flint, Michigan water crisis was one publicized example of contamination from failed Pb pipes [3], and recent evaluations of US drinking water revealed other cities with similar problems [4]. In low- and middle-income countries (LMICs), Pb exposure occurs through various contamination sources including Pb-acid battery production and recycling, Pb-glazed ceramics, and limited uses of leaded-gasoline [5,6].

While the scientific consensus is that there is no safe level of Pb, a considerable amount of research has been devoted to understanding the impact of Pb exposure at or around the blood Pb reference value of 5 µg/dL, especially in the early stages of development [7]. Pregnant women and children are among the most vulnerable to Pb exposure, which is known to be especially toxic during periods of rapid growth [8], and it has been postulated that epigenetic perturbations by perinatal environmental exposures such as Pb could have persistent health effects throughout life [9]. For children, early life exposure can occur in utero via placental transfer, in infancy through breast milk, and directly from the drinking water and other sources (e.g., soil, peeling paint or Pb dust in older homes) throughout childhood. Such exposures may disproportionately affect susceptible populations, including pregnant women, children, and racial or ethnic minority groups that experience disparities in access to healthy housing or clean drinking water [10].

At the interface between environment and inherited genetics lies epigenetics—a suite of molecular mechanisms that give rise to a pseudo-Lamarckian [11] intra-organism adaptability. Epigenetics is the study of chemical modifications to DNA or chromatin, which can be passed from one cell generation to the next influencing gene expression without altering the underlying DNA sequence, and with such modifications remaining potentially reversible throughout the life-course [12]. Although epigenetic modifications are natural and necessary, aberrant patterns at both the gene and genome-wide level have been linked to many chronic health conditions including metabolic diseases [13], cancer [14], and Alzheimer’s disease [15]. Epigenetics is thought to be one biological mechanism underlying the developmental origins of health and disease (DOHaD) hypothesis, which postulates that early-life exposures can impact later-life health outcomes [16]. Accordingly, toxicant exposure during periods of rapid growth (e.g., embryogenesis), when the epigenome is being patterned and fidelity is paramount, could cause deregulation of gene expression and increase the risk of disease [17].

The toxicity of Pb is best known in the context of neurological-related outcomes, but data from recent animal and human studies corroborate the DOHaD theory and suggest a potentially broader impact of Pb exposure with sex-specific associations with other health outcomes such as adiposity and cardiometabolic risk in animal [18,19] and human studies [20,21]. These long-lasting, wide-spread effects of Pb exposure could be explained in part by the influence of Pb on epigenetic marks. Increasingly, evidence suggests gestational Pb exposure influences offspring epigenetic programming in mouse and human studies utilizing candidate-gene [18,22,23] or epigenome-wide approaches [8,24,25,26] to assess DNA methylation. Epigenome-wide DNA methylation platforms can help to identify candidate gene pathways that shed light on the impact of Pb on development processes and later-in-life risk for neurological, cardiovascular, or other adverse health outcomes.

Understanding the early-life exposome and the short- and long-term impact it exerts on offspring health hinges on the availability of relevant exposure and outcome data. In a traditional cohort study model, this would be limited to the number of participants recruited, successfully sampled and followed prospectively. Neonatal bloodspots offer a unique opportunity in that the sample is collected from nearly all children at birth in many countries, with the caveats that not all states/countries retain bloodspots, some parents refuse collection or storage, and some storage programs may have started after the population of interest was born. As of 2011, 20 states in the US retained bloodspots to be used for research purposes, though procurement and storage processes vary widely between states [27]. Increasing interest in genetic and epigenetic studies utilizing bloodspots has generated much controversy, and guidance for researchers have been discussed in this context for the US [28] and globally [29]. Despite these difficulties, bloodspots represent a unique and precious resource that needs to be handled with care, both physically and ethically. Our study is just one example of the broad applications bloodspots have in expanding our understanding related to environmental influence on health and disease.

In this study, we hypothesize that gestational Pb exposure will be associated with newborn DNA methylation profiles and that we can assess both Pb levels and DNA methylation via neonatal bloodspots. To test this hypothesis, we leveraged a study of mother–child pairs with archived neonatal bloodspots, the Healthy Families (HF) Project [30]. The HF study has been leveraged previously to study gene-specific epigenetics in relation to health status [31]; however, here we used archived bloodspots to study at-birth epigenome-wide profiles among the participants in relation to gestational toxicant exposure. Specifically, we measured Pb in neonatal bloodspots as a proxy measure of late gestational exposure and measured genome-wide DNA methylation via the Infinium MethylationEPIC array in the same bloodspots.

## 2. Materials and Methods

### 2.1. Healthy Families Project

In order to examine health outcomes in children across sensitive developmental periods, the HF project recruited 40 families with children 12–24 months old (toddlers), 40 families with children 3–5.99 years old (preschool), and 52 families with children 10–12.99 years old (school-aged) within 1-h driving distance to Ann Arbor, Michigan, US. During a home visit by a research assistant, families provided survey data, child and parent anthropometric measurements, and written, informed consent. Questionnaires were used to assess sociodemographic characteristics, which included measures of maternal education, child race/ethnicity, and child sex. Methods and details on the original study were previously detailed [31]. While the HF project originally focused on health and behavior of children and their families, we leveraged data and archived samples from this study population to estimate gestational exposure to Pb and assess its impact on the child’s epigenome at birth. To do so, we selected a subset of HF participants from all recruitment age-groups with archived neonatal bloodspots for inclusion (Figure 1).

### 2.2. Neonatal Bloodspots

The US state of Michigan has collected and archived neonatal bloodspot samples from most children born in the state since 1984. These samples are collected via heel prick in the hospital within 24 h of birth and are used for genetic screening of rare disorders. The remaining bloodspots are stored at the Michigan Neonatal Biobank where they can be accessed with proper consent. For recruited participants with neonatal bloodspots available (see Figure 1), specimens were sourced from the Michigan Neonatal Biobank. Consent was obtained from all recruited families prior to bloodspot procurement. Michigan Department of Health and Human Services (MDHHS) and University of Michigan (UM) IRB approvals were obtained for the Healthy Families project (UM: HUM00079730) and DNA isolations from neonatal bloodspots (MDHHS: 201311–04-EA). Before retrieval, Michigan Neonatal Biobank bloodspots were stored at different temperatures depending upon their collection date. As such, the current age of each recruited child corresponded to the storage method for their bloodspot: toddler and preschooler bloodspots were stored at −20 °C, school-aged children’s bloodspots were stored at 20 °C. Upon receiving the bloodspots for use in epigenetic analyses, they were stored in sealed bags at 4 °C.

### 2.3. Bloodspot Pb Assessment

All supplies were acid-cleaned, and bloodspot samples were processed and analyzed in Class 100 and 1000 ultraclean rooms at the Michigan State University Exposure Science Laboratory. Two bloodspot punches per sample were excised using a 3-mm sterile biopsy punch. In order to assess background levels of Pb in blank filter card material, two blank 3-mm punches next to each bloodspot were also excised. Each 3-mm bloodspot punch was assumed to contain 3.1 μL of blood based on previous studies [32,33]. Each punched sample was transferred to an acid-cleaned centrifuge tube using Teflon-coated tweezers, and 3 mL of extraction solution (5% ultrapure grade HNO3, and 0.05% Triton X-100 in Type I water) was added. The solutions were centrifuged at 3600× *g* (6200 rpm) for 2 min and then incubated for 90 min at room temperature on a shaker table at 300 rpm. Extracts were then analyzed for Pb using high-resolution inductively coupled plasma-mass spectrometry (HR-ICP-MS, ELEMENT2, Thermo Finnigan, San Jose, CA, USA). Five-point calibration curves were created using Pb standards (SPEX CertiPrep, Metuchen, NJ, USA) in 5% nitric acid solution. National Institute of Standards and Technology (NIST, Gaithersburg, MD, USA) SRM 1640a was used as an external quality control standard to check the calibration of the instrument, and NIST SRM 955c caprine blood was used to assess the recovery and accuracy of the sample digestion and analysis by spiking blank Whatman #903 Protein Saver cards. This analysis method incorporates daily quality assurance and quality control measures including field blanks, Type I water blanks, replicate analyses and external standards as described elsewhere [34,35]. The method detection limit (MDL) for Pb was determined to be 0.07 µg/dL.

### 2.4. DNA Isolation

DNA was isolated from neonatal bloodspots using a modified version of the QIAamp DNA Micro Kit (Qiagen, Cat. #56304, Hilden, Germany). To maximize DNA yields from bloodspots, a number of changes were made to the standard QIAamp DNA Micro Kit protocol. First, isolations were performed on four 3mm bloodspot punches rather than the recommended three punches. Second, to ensure complete digestion, the heated incubation step was increased from 1 h to 2 h. Third, the digested solution containing lysis buffer and bloodspot punch material were transferred to a spin basket (Qiagen Cat# 19598) over a collection tube before first spin. DNA was eluted with nucleic acid free water heated to 70 °C in a two-step process, first with 45 µL followed by 30 µL, each with a 5-min incubation before spinning. Across all bloodspot samples (*n* = 129), average DNA yield (SD) was 2.1 µg DNA (SD = 0.6 µg).

### 2.5. MethylationEPIC Analysis

Quality control and DNA methylation analysis via the Infinium MethylationEPIC occurred at the University of Michigan Epigenomics and Advanced Genomics Cores following standard procedures. Briefly, the Qubit high sensitivity double stranded DNA assay was used to quantify DNA. For a subset of 96 samples (see Figure 1), 250 ng were bisulfite converted with the EZ DNA Methylation kit (Zymo) using the manufacturer’s incubation parameters specific for Illumina arrays. Samples were hybridized to the Infinium MethylationEPIC BeadChip and scanned at the Advanced Genomics Core. Raw image files were then read into R using the minfi Bioconductor package [36]. Quality control was performed using the Enmix Bioconductor package. Quality control procedures include examining data for outlier samples, (with low intensity overall and/or by overall distribution of methylation across all probes), removing poorly performing samples (with estimated sex mismatch or with >5% of probes with low detection), and removing poorly performing probes (detection *p*-value > 0.05 in at least 5% of samples [37]. Additionally, probes from the X and Y chromosomes, cross-reactive probes, and probes with polymorphisms in the CpG site were excluded. All samples passed quality control, and 789,147 probes were retained for analysis. Probe intensities were corrected for background and dye bias [36]. Probes were annotated using the ilm10b4.h19 annotation R package, which includes genomic locations and features (gene name, regulatory regions, evidence for each locus being in an enhancer, DNAseI hypersensity site, etc.).

### 2.6. Statistical Methods

All statistical analyses were performed in the R Software for Statistical Computing. We first calculated descriptive statistics on neonatal bloodspot Pb concentrations and all covariates, and assessed distributions of continuous variables. We estimated cell-type proportions from each sample using an algorithm based on Infinium data and a reference dataset from sorted cord blood cells [38,39]. We performed analyses (Spearman correlations and ANOVA tests) to identify covariates associated with Pb (i.e., potential confounders) from among: technical factors (EPIC chip and chip position), estimated cord blood cell-type proportions (nucleated red blood cells (nRBCs), CD4+ T-cells, CD8+ T-cells, B cells, monocytes, granulocytes, and natural killer cells), maternal (BMI, race [white vs. all other], education) and child characteristics (sex, age group). None were significantly associated with neonatal Pb levels at *p* < 0.05. We next used the R package ChAMP to perform singular value decomposition to identify technical or biological covariates associated with principal components of the entire DNA methylation dataset [40]. In this analysis, a Kruskal–Wallis test for categorical covariates or a linear model for continuous covariates were used to test associations between covariates and each of the principal components of the DNA methylation beta matrix. Race, child’s sex, recruitment group, and cell type estimates were associated with the first four principal components of DNA methylation.

We used least squares regression to identify differentially methylated CpG sites (using beta values for each of 789,147 CpG sites as the outcomes) associated with Pb exposure. We used an empirical Bayes model to moderate the standard errors and determine a significance value for each model in the limma package [41]. While none of the available covariates had evidence for confounding (associated with both Pb and DNA methylation data), we included sex, race, and estimated cell-type proportions (granulocytes, nucleated red blood cells, B cells and monocytes) as covariates in the final models due to their highly significant associations with DNA methylation profiles in this study. We did not include recruitment group as we found it to be confounded by cell-type proportions (Appendix A). We also report results for a model excluding cell type estimates adjusting only for sex and race. The association with Pb was tested first using neonatal Pb as a continuous variable (in µg/dL). In the continuous Pb analysis, values below the MDL were set to MDL/sqrt (2). Since 16% of the values were below the MDL, we also tested the association with Pb as a dichotomous variable, splitting samples equally into two groups (above and below the median of 0.55 µg/dL).

We considered *p* < 9 × 10^−8^, the p-value recommended for MethylationEPIC studies based on the number of multiple comparisons taking into account the correlation between some loci on the array [42], as the strongest evidence for a statistically significant association between Pb and DNA methylation at each locus. Given our small sample size and limited statistical power, we also discuss results as potential avenues for future exploration with false discovery rate q-value < 0.2. In addition to site-specific analyses, we identified differentially methylated regions (DMRs; regions consisting of two or more neighboring CpG sites that were associated with Pb exposure, using an algorithm called DMRcate [43]. In the DMRcate analysis, the Gaussian kernel bandwidth was set to 1000 with a smoothing function of C = 2. We also assessed whether any biological pathways were enriched amongst the top 10,000 CpG sites (by raw p-values ranging from 1.33 × 10^−6^ to 0.014; [44]). For the association with continuous and categorical Pb measures using a gene ontology enrichment algorithm designed for Infinium data that adjusts for the number of CpG sites within each gene represented on the array [44]. This analysis was run for Gene Ontology (GO) and KEGG pathways, and results with an FDR adjusted value of q < 0.2 are discussed.

Whereas the methods listed above assess the influence of Pb on increases or decreases DNA methylation at a given site, toxicant-induced increases in DNA methylation variability have also been hypothesized [45]. Thus, we utilized DiffVar, a function in the missMethyl Bioconductor R package to test for Pb-related changes in DNA methylation variability. This method is based off of Levene’s z-test, and involves performing a *t*-test on the absolute deviations of the M-values (logit-transformed beta values) from the group mean to test the null hypothesis that group variances are equal. An empirical Bayes framework is then used to moderate the t-statistics prior to significance calling. This method is robust against non-normality and outliers [44,46]. We performed this analysis using a model adjusting for cell-type proportions, sex, and race.

## 3. Results

### 3.1. Cohort Characteristics

Mothers recruited to the HF study (*n* = 129) were 77% white and had an average BMI of 29.3 kg/m^2^ ± 8.8 standard deviation. The children who participated were 54% male and the proportions across the three age groups of toddlers, preschool-age and school-aged were 30%, 26% and 44%, respectively (see Table 1). Children and their mothers from the subset selected for EPIC analysis were not significantly different from families from the HF study overall.

### 3.2. Sample Characteristics

This study utilized neonatal dried bloodspots that were archived by the State of Michigan and underwent different storage conditions and storage time depending on the participant recruitment group. In group 1 (*n* = 40), samples were stored at −20 °C for approximately 5 years (see Figure 1). In group 2 (*n* = 39), samples were stored at −20 °C for approximately 8 years. In group 3 (*n* = 50), samples were stored at room temperature for approximately 16 years. Despite the varying storage conditions, we demonstrated that DNA of adequate quality and quantity could be extracted from four 3-mm punches, undergo bisulfite conversion, and pass all quality control checks included in the Infinium MethylationEPIC pipeline for all three groups. When completing the singular value decomposition (SVD) analysis, we noted that the sample group was associated with principal components 1, 3, and 7 of the DNA methylation data (*p* < 0.05; Appendix A). In bivariate analysis, we noted that the association between the recruitment group and DNA methylation was largely confounded by cell-type distributions. Appendix A displays the mean estimated cord blood cell-type proportions by the recruitment group. In the third group of children ages 10–12 years, estimated proportions of nRBCs and CD8+ T-cells were significantly higher than in samples from the other two age groups (ANOVA *p* < 0.01). Since cell-type proportions are a known strong predictor of variation in DNA methylation, we included cell-type estimates instead of the sample group in final statistical models.

### 3.3. Pb Assessment in Dried Bloodspots

Pb levels were detectable in all bloodspot samples and blank filter card samples. Exposure levels reported in this paper were calculated by subtracting Pb levels of each blank filter card sample from its corresponding bloodspot sample. The average Pb concentrations in blank filter card samples and in uncorrected bloodspot samples were 1.31 ± 0.59 µg/dL and 2.25 ± 1.02 µg/dL, respectively. After blank correction, bloodspot sample Pb concentrations ranged from <MDL to 5.27 µg/dL with a median concentration of 0.55 µg/dL. Twenty-one corrected samples (16%) were below the MDL.

### 3.4. CpG Site Loci and Regions Differentially Methylated by Pb

In the site-specific analysis of 789,147 CpG sites, we identified a number of genes with differential methylation by levels of Pb exposure. In the model without cell-type adjustment and including Pb concentration as a continuous variable, we observed 30 CpG sites that showed decreasing methylation and 3 CpG sites that showed increasing methylation with increasing Pb exposure. For this analysis, we used a p-value cutoff recommended for epigenome-wide studies that use the MethylationEPIC array (*p* < 9 × 10^−8^; Appendix A). Given the small sample size, we also report CpG sites significant at an FDR q-value < 0.2 for the exploration of potential associations with exposure in future studies. In the same model without cell-type adjustment, we observed 99 and 23 CpG sites that showed decreasing and increasing methylation, respectively, with Pb exposure at a relaxed q < 0.2. Out of these sites, 66% were in DNAseI hypersensitivity sites, 9% were in transcription factor binding sites, 11.5% were in CpG islands, and 18% were in suspected or known enhancers.

Given the broad influence of cell-type distribution on DNA methylation profiles, we next adjusted for cell-type proportions and did not observe any CpG sites associated with Pb as a continuous variable at *p* < 9 × 10^−8^. Table 2 displays results for the 32 CpG sites associated with Pb at q < 0.2 (27 decreasing and 5 increasing) after adjusting for cell types. Many of these sites were in at least one known or expected regulatory region including six (19%) in CpG islands, three (9%) in transcription factor binding sites, 19 (59%) in DNAseI hypersensitivity sites, and seven (22%) in enhancers. Thirty of the thirty-two loci in the cell-adjusted model were also significant in the cell-free model.

We utilized DMRcate to identify regions with at least two neighboring CpG sites that were associated with Pb in the same direction. We identified four regions with decreasing methylation at 3, 4, 8, and 17 CpG sites and two with increasing methylation at four CpG sites each (Table 3).

Since 16% of bloodspot samples fell below the MDL, we also assessed associations with DNA methylation after categorizing Pb exposure. When dichotomizing Pb exposure into two groups by median concentration (0.55 µg/dL), no CpG sites were significantly associated with the Pb group at *p* < 9 × 10^−8^ or at q < 0.2, whether or not cell proportions were included in the model. Based on these results, we did not conduct DMR analysis for dichotomized Pb.

### 3.5. Biological Pathways Enriched for Differential Methylation by Pb

Six GO pathways were enriched amongst the 10,000 CpG sites with the lowest p-value for their association with continuous Pb exposure (Table 4). Amongst these, four pathways involved in cell morphogenesis and cellular adhesion were also enriched in the dichotomous Pb analysis. In total, 13 GO and two KEGG pathways were enriched amongst the top 10,000 CpG sites by dichotomous Pb. These pathways included morphogenesis for cells and neuron projections, post-synapse organization, cellular adhesion, axon guidance, and GABAergic synapses.

### 3.6. Differential Variability by Pb Exposure

Exposure-induced changes that result in altered epigenetic programming or maintenance may manifest as changes in mean methylation (i.e., presented in the differential methylation analysis), or could instead result in more variability from increased stochasticity. With this in mind, we conducted “DiffVar” analysis, an approach based on Levene’s z-test, to identify CpG sites with increasing or decreasing variability with more Pb exposure. While no differentially variable CpG sites were detected when modeling Pb as a dichotomous variable, 16 CpG sites had higher variability as continuous Pb levels increased (q < 0.2), including two CpG sites at q < 0.05 after adjusting for cell-type proportions, sex, and race (Appendix A).

## 4. Discussion

In this study, we leveraged the State of Michigan’s Neonatal Biobank to access archived bloodspot specimens and measure at-birth exposure to Pb as well as epigenetic patterns. While nearly all at-birth specimens had a Pb measurement lower than the US CDC’s reference value of 5 µg/dL, we detected differential DNA methylation by Pb exposure including increased variability at some CpG sites and decreased or increased DNA methylation by exposure at others. The average and range of the Pb levels measured here (average 0.78 µg/dL bloodspot Pb with a maximum of 5.27 µg/dL) are similar to levels reported in the US according to the most recent National Health and Nutrition Examination Survey (NHANES) [47], providing confidence in our technique. While this study is one of only a few to show this in the context of bloodspot Pb levels, our data add to a growing body of evidence which suggests Pb can influence epigenetic patterns. For example, the ELEMENT cohort study (*n* = 247) has linked early life Pb exposure, which was measured in maternal bone, maternal blood, and cord blood, with early life and peri-adolescent DNA methylation using candidate-gene methods [23,48,49]. As a second example, Project Viva (*n* = 268), a US-based prebirth cohort, has also demonstrated that maternal blood Pb levels during pregnancy are related to cord blood epigenome-wide DNA methylation patterns. In contrast to the higher average in the Mexico-based ELEMENT study, Project Viva recorded an average erythrocyte Pb level of 1.22 µg/dL ranging from 0.29–4.97, which is more in line with our participant exposures [26]. An additional Mexico City study showed that prenatal Pb (in addition to multiple other metals) impacts cord blood DNA methylation at genes related to DNA repair [50]. Complementing this human data, animal studies support the toxicoepigenetic impact of Pb. Perinatal Pb exposure in mice via maternal drinking water has been shown to impact DNA methylation patterns at retrotransposons in the brain [22], metastable epialleles in tail tip DNA [18] and promoters throughout the genome in DNA from sorted cortical neurons [51]. These findings are consistent with other reports that that early life Pb is related to murine DNA methylation patterns, and that the epigenetic effects of Pb exposure may persist throughout life [52].

The differentially methylated genes in our study showed enrichment for biological pathways, including cellular organization, neurodevelopment, and neurological function. In some ways, the latter may be unsurprising given the known link between Pb exposure and neurodevelopment, but identifying these relationships in infant bloodspots further bolsters the potential utility of this specimen and highlights the potential impact of relatively low blood Pb levels during development. Cellular organization terms were also common in a study that evaluated the hippocampal methylome of rats developmentally exposed to Pb, in which the authors showed that rats in the lowest exposure group had the most overrepresented GO terms compared to the control group [53]. This non-monotonic relationship highlights the need to study low-level exposures. In addition to GO terms found in our study, the identified differentially methylated CpG sites according to Pb levels on the MethylationEPIC chip represent potential future avenues of research. For example, we showed decreased methylation at a CpG site annotated to *PRDM16* in Table 2. This same gene was recently identified in an epigenome-wide study of Parkinson’s disease [54], which suggested epigenetic programming may contribute to this disease and that postulated environmental exposures could induce the programming. Taken together with our findings, *PRDM16* could be a candidate gene for studies evaluating the long-term impact of early life metal exposure on neurodegeneration.

In addition to differential methylation, we identified sites of epigenomic variability that were associated with early life Pb exposure. We previously advanced this concept in a contemporary review [45]. The premise is that exposure can result in aberration to include deregulation of methylation and increased variability, with the latter resulting from increased stochasticity. We coined the phrase “environmental deflection” to denote the exposure-induced change from a normal epigenetic profile over time as well as the possible increased vulnerability of younger organisms to exposure relative to older. The list of genes identified in our study as more variably methylated in relation to increasing Pb exposure may be interesting targets for future research. For example, Ferredoxin 1 (*FDX1*) is involved in adrenal steroidogenesis as well as bile acid and vitamin D synthesis [55]. The dysregulation of iron–sulfur cluster genes, like *FDX1*, are known to cause human disease (e.g., infant multiple mitochondrial malfunctions syndrome and infant multiple mitochondrial malfunctions syndrome [56]) and thus exposure-induced epigenetic variability at this locus warrants further study.

In this study, we utilized bloodspots to gain valuable insights into the early life environment and its influence on the at-birth epigenome in a study population who were 1–13 years of age at recruitment. Bloodspots have been successfully used to determine early life exposures including Vitamin D [57], perfluorinated chemicals [58], and endocrine-disrupting chemicals [59]. Similarly, bloodspots have been used to evaluate outcome measures such as immune function [60,61], thyroid hormones [62] and insulin concentration [63]. Epigenome-wide and targeted patterns have also been assessed recently using bloodspots. Gao et al. conducted an epigenome-wide analysis using the HumanMethylation 450 array (a predecessor to the MethylationEPIC array) on infant bloodspot specimens and identified a relationship between the epigenetic pattern of a gene known to play a role in asthma and the risk of wheeze by age six [64]. Another group evaluated bloodspot DNA methylation profiles using the HumanMethylation 450 array and found evidence that gestational folic acid deficiency may influence risk for cleft palate through an epigenetic mechanism [65]. The HF study children have been previously studied, and bloodspots were used to assess gene-specific DNA methylation (via pyrosequencing) and relate at-birth profiles with childhood weight status [31]. The use of archived samples, both in the present study and in the research mentioned above, allows for a unique retrospective exposure and epigenetic assessment in newly recruited study participants.

There is growing interest in metal exposure assessment via bloodspots in various environmental health and public health studies due to the relatively non-invasive nature of blood sample collection. For example, Basu and colleagues at McGill University have published on mercury assessment using bloodspots in both wildlife [66] and human infant samples [32,67] and Ruden and colleagues at Wayne State University have published on Pb assessment as well as DNA methylation analysis using bloodspots. In line with our findings, Sen et al. demonstrated in a cohort of 45 children that Pb levels in early childhood are associated with differential DNA methylation patterns at that same timepoint [25]. This same group has also investigated the multigenerational influence of at-birth Pb levels on the epigenome using bloodspots from mother–infant pairs [68]. Combined with our results, these data suggest that developmental Pb exposure can impact the programming of DNA modifications early in life. It should be noted however that the precision and accuracy of bloodspot sample analysis are still being evaluated [69,70,71]. For example, Funk et al. reported that blank blood collection cards contained significant Pb concentrations and suggested that blank collection cards be cleaned before their use [72]. Our Pb bloodspot data also support their findings in that more accurate measurements can be obtained by accounting for the Pb content of blank collection cards.

While leveraging archived bloodspots has the potential to accelerate the discovery of epigenetic biomarkers and advance environmental epigenetic research, storage conditions (e.g., time and temperature) may impact results. A recent study compared data quality and results for associations between smoking or age and DNA methylation from the Infinium 450 array, run with matched samples from 62 adults. One sample was stored in an EDTA-tube before DNA extraction while the other sample was spotted on a Whatman FTA card and stored at room temperature for 5–10 years before extraction. Importantly, DNA methylation profiles from the two sample types were highly correlated and produced similar results for the association studies without moderation by storage type [73]. Whether neonatal bloodspot samples are also robust against storage conditions, including storage for longer than 10 years, is in need of full assessment. In our study, estimated proportions of CD8+ T-cells and nRBCs were significantly higher in samples from the oldest recruited group; these samples had the longest time in storage prior to DNA isolation and storage occurred at room temperature instead of in a freezer. Whether these differences in storage conditions impacted the amount of DNA that survived from various cell types should be tested experimentally. In the meantime, adjustment for cell-type proportions is imperative when working with epigenetic data from bloodspots.

Our study highlights the utility of infant bloodspots and adds to a growing number of studies using this specimen to assess at-birth exposures. However, we do acknowledge several limitations. First, while we leveraged bloodspots as a way to retrospectively sample participants, future studies will need to consider tissue from target tissues. Second, due to the small sample size, we had limited statistical power to detect all true associations between Pb and DNA methylation, especially at CpG sites with small effect sizes. We were also unable to perform sex-stratified analyses due to sample size. Third, results from the present analysis may not be generalizable to populations with different levels of exposure or racial/ethnic distributions. Finally, we recognize that the cross-sectional nature of the exposure and DNA methylation measurements represent a unique strategy in the context of bloodspots, but future studies should consider serial blood (or target tissue) assessment to more fully characterize the exposure to Pb and its impact on DNA methylation profiles over time (incidence and persistence of alteration).

## 5. Conclusions

In this US-based study, we leveraged dried bloodspots from birth to report on neonatal Pb exposure and capillary blood leukocyte DNA methylation. While our cohort’s median bloodspot Pb levels were relatively low, we were able to identify associations between DNA methylation and Pb at 32 CpG sites, with the majority being inversely associated with Pb concentrations. Our pathway analysis suggests that low-level Pb exposure is related to development and neurological function. We also demonstrate further evidence of phenomena we call environmental deflection in which developmental exposure induces not only altered average methylation but also increased variability among those with higher Pb exposure. This study also serves to highlight the utility of bloodspots as a source of at-birth exposure and outcome data. By combining thoughtful collection and storage of bloodspot specimens by local authorities and ethical stewardship by researchers, environmental epigenetic studies around the world that aim to better understand exposure-induced disease would be enhanced with this unique biological sample.

## Figures and Tables

**Figure 1 ijerph-17-06775-f001:**
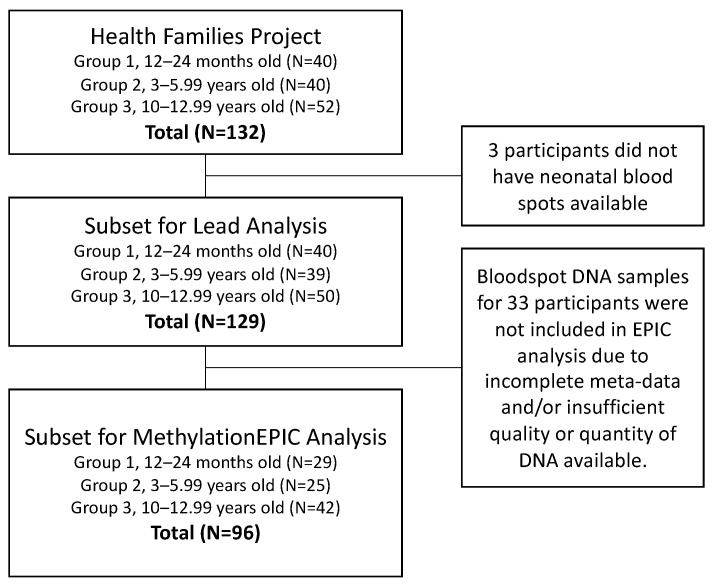
Participant inclusion flowchart for Pb analysis and DNA methylation analysis.

**Table 1 ijerph-17-06775-t001:** Cohort Characteristics and Pb Exposure Assessment.

	Included in MethylationEPIC Analysis, *n* = 96		All HF Families, *n* = 129	
Characteristics	Mean ± SD or N (%)	Range	Mean ± SD or N (%)	Range
*Mothers*				
BMI (kg/m^2^)	29.3 ± 8.9	15.4–66.4	29.5 ± 8.8	15.3–66.4
Obesity				
Not obese	60 (63%)		77 (60%)	
Obese	34 (35%)		55 (39%)	
Race				
Non-white	22 (23%)		29 (22%)	
White	74 (77%)		99 (77%)	
*Children*				
Male (%)	52 (54%)		69 (53%)	
Recruitment Group				
Group 1	29 (30%)		40 (31%)	
Group 2	25 (26%)		39 (30%)	
Group 3	42 (44%)		50 (39%)	
*Estimated Cord Blood* *Cell Type %*				
Nucleated Red Blood Cells (nRBC)	4.2 ± 2.0	0.5–12.8	na	na
CD8+ T cells	7.5 ± 3.4	2.3–23.6	na	na
CD4+ T cells	19.3 ± 6.5	6.6–43.5	na	na
B cells	6.1 ± 2.8	1.4–21.0	na	na
Monocytes	9.0 ± 2.3	4.1–17.9	na	na
Granulocytes	57.0 ± 10.4	17.5–77.1	na	na
Natural Killer Cells	0.01 ± 0.1	0–0.96	na	na
*Pb Exposure Assessment*				
Blood spot Pb (µg/dL)	0.78 ± 0.85	<MDL–5.27	0.93 ± 0.80	<MDL–5.27
Pb Category				
Low (<median 0.55 µg/dL)	48 (50%)		69 (53%)	
High (≥median 0.55 µg/dL)	48 (50%)		60 (47%)	

Note, 2 moms are missing BMI. In the full study sample, 128 have data on race (1 is missing). SD, standard deviation; na, not applicable; MDL, method detection limit. Group 1, children 12–24 months old at time of recruitment; Group 2, children 3–5.99 years old; and Group 3, children 10–12.99 years old.

**Table 2 ijerph-17-06775-t002:** Differentially Methylated CpG Sites by Pb (q < 0.2).

Probe ID	Position	Gene Name	Relationship to CpG Island	Estimate (per ug/dL Pb)	SE of Estimate	Average % Methylation at the CpG Site	Raw *p*-Value	q-Value
cg03744954	chr7:23637556	*CCDC126*	Island	0.010	0.002	2.2%	1.33 × 10^−6^	0.12
cg11961702	chr10:1130138	*WDR37*	Open Sea	−0.007	0.001	98.1%	1.08 × 10^−6^	0.12
cg09489281	chr5:43604149	*NNT*	S. Shore	0.027	0.005	5.1%	1.19 × 10^−6^	0.12
cg06157837	chr16:50130934	*HEATR3*	Open Sea	−0.021	0.004	92.7%	1.08 × 10^−6^	0.12
cg12819470	chr7:18013541		Open Sea	−0.022	0.004	94.1%	1.34 × 10^−6^	0.12
cg00694932	chr22:49029825	*FAM19A5*	Open Sea	−0.027	0.005	87.6%	4.91 × 10^−7^	0.12
cg12666827	chr17:81043176	*METRNL*	Island	−0.026	0.005	92.1%	5.19 × 10^−7^	0.12
cg17533118	chr2:204664739		Open Sea	−0.030	0.006	88.3%	8.35 × 10^−7^	0.12
cg16393928	chr1:3135836	*PRDM16*	Open Sea	−0.024	0.005	90.8%	1.46 × 10^−6^	0.12
cg27500206	chr11:4600140	*C11orf40*	Open Sea	−0.031	0.006	90.8%	7.93 × 10^−7^	0.12
cg25968149	chr3:174255312		Open Sea	−0.020	0.004	92.9%	1.80 × 10^−6^	0.12
cg20383654	chr1:39490535	*NDUFS5*	Open Sea	−0.018	0.004	93.0%	1.82 × 10^−6^	0.12
cg05355328	chr19:33096524	*ANKRD27*	Island	−0.022	0.004	94.2%	1.98 × 10^−6^	0.12
cg25104648	chr18:18566146	*ROCK1*	Open Sea	−0.022	0.004	95.0%	2.57 × 10^−6^	0.13
cg08236836	chr2:95613608		Open Sea	−0.017	0.003	98.4%	2.54 × 10^−6^	0.13
cg25198485	chr4:58086247		Open Sea	−0.021	0.004	92.2%	2.46 × 10^−6^	0.13
cg23724374	chr6:150973466	*PLEKHG1*	Open Sea	−0.019	0.004	89.7%	3.37 × 10^−6^	0.15
cg11877273	chr1:29182535	*OPRD1*	Open Sea	−0.019	0.004	94.1%	3.61 × 10^−6^	0.15
cg06019448	chr4:190417346		Open Sea	−0.023	0.005	90.5%	3.53 × 10^−6^	0.15
cg06051201	chr1:53925306	*DMRTB1*	Island	−0.026	0.005	95.1%	4.37 × 10^−6^	0.17
cg09980056	chr6:52366854	*TRAM2*	Open Sea	−0.025	0.005	95.0%	4.60 × 10^−6^	0.17
cg09413029	chr10:97152467	*SORBS1*	Open Sea	−0.022	0.005	93.0%	4.49 × 10^−6^	0.17
cg14810301	chr5:88169099	*MEF2C*	Open Sea	−0.018	0.004	93.3%	5.41 × 10^−6^	0.17
cg15406566	chr16:4606279	*C16orf96*	Open Sea	0.011	0.002	87.4%	5.55 × 10^−6^	0.17
cg05026437	chr3:99527476	*MIR548G*	Open Sea	−0.021	0.004	92.5%	5.44 × 10^−6^	0.17
cg14645844	chr8:123831620	*ZHX2*	Open Sea	-0.024	0.005	93.7%	5.17 × 10^−6^	0.17
cg10174926	chr14:72818968	*RGS6*	Open Sea	−0.018	0.004	93.0%	5.81 × 10^−6^	0.17
cg07431286	chr5:72526496		Island	0.003	0.001	3.1%	6.04 × 10^−6^	0.17
cg27094173	chr11:73371753	*PLEKHB1*	N. Shore	−0.008	0.002	92.9%	7.49 × 10^−6^	0.20
cg00394874	chr6:30747276		Open Sea	0.048	0.010	25.4%	7.85 × 10^−6^	0.20
cg06534673	chr16:1210311	*CACNA1H*	Island	−0.027	0.006	71.4%	7.73 × 10^−6^	0.20
cg07366047	chr2:135921205	*RAB3GAP1*	Open Sea	−0.029	0.006	94.1%	7.97 × 10^−6^	0.20

**Table 3 ijerph-17-06775-t003:** Differentially Methylated Regions (DMRs) by Pb Exposure.

Position	Genes Overlapping This Region	# of CpG Sites	Mean Difference in Proportion Methylated per 1 µg/dL Pb	Min. Smoothed FDR q-Value
chr19:33096524-33096688	*ANKRD27*	3	−0.006	3.15 × 10^−6^
chr1:53925145-53925368	*snoU13*, *Y_RNA*, *SCARNA16*, *U1*, *SCARNA18*, *SCARNA24*, *SNORD112*, *SNORA62*, *SNORA63*, *SNORD46*, *DMRTB1*, *SNORA2*, *SNORD81*, *U3*, *SNORA51*, *SNORA25*, *SCARNA20*, *SNORA67*, *U6*, *SNORA70*, *SNORA77*, *SNORA26*, *U8*, *SCARNA11*, *SNORA31*, *SNORA42*, *SNORA40*, *SNORD64*, *ACA64*, *snoU109*, *SNORD60*, *SNORD116*	4	−0.020	2.12 × 10^−6^
chr9:123605229-123605666	*PSMD5-AS1*, *PSMD5*	8	−0.015	5.42 × 10^−7^
chr15:93616424-93617402	*RGMA*	17	−0.014	5.60 × 10^−14^
chr11:111637306-111637615		4	0.010	3.06 × 10^−5^
chr17:6796745-6797034	*ALOX12P2*, *AC027763.2*	4	0.016	6.21 × 10^−5^

These results are from the model adjusting for sex and race, with Pb as a continuous variable. Models adjusting for cell type did not have any significant DMRs. FDR, false discovery rate.

**Table 4 ijerph-17-06775-t004:** Enriched Biological Pathways by Top 10,000 Genes by p-value in Cell-type adjusted models.

	Pathway Identifier	Description of Pathway	# of Genes in Pathway	# Differentially Methylated Genes	*p*-Value	q-Value	# Differentially Methylated Genes	*p*-Value	q-Value
GO Analysis—Dichotomized Pb							GO—Continuous Pb		
GO: BP	GO:0000902	cell morphogenesis	990	390	1.48 × 10^−6^	0.025	385	2.65 × 10^−5^	0.139
GO: BP	GO:0000904	cell morphogenesis involved in differentiation	709	292	2.24 × 10^−6^	0.025			ns
GO: BP	GO:0022610	biological adhesion	1369	467	5.47 × 10^−6^	0.034	472	1.23 × 10^−5^	0.131
GO: BP	GO:0007155	cell adhesion	1363	465	5.93 × 10^−6^	0.034	469	1.73 × 10^−5^	0.131
GO: BP	GO:0032989	cellular component morphogenesis	1098	417	1.03 × 10^−5^	0.047	421	9.31 × 10^−6^	0.131
GO: BP	GO:0032990	cell part morphogenesis	653	269	1.25 × 10^−5^	0.047			ns
GO: BP	GO:0048812	neuron projection morphogenesis	617	256	1.83 × 10^−5^	0.058			ns
GO: BP	GO:0120039	plasma membrane bounded cell projection morphogenesis	631	261	2.10 × 10^−5^	0.058			ns
GO: BP	GO:0048858	cell projection morphogenesis	635	262	2.30 × 10^−5^	0.058			ns
GO: BP	GO:0048667	cell morphogenesis involved in neuron differentiation	555	234	3.05 × 10^−5^	0.069			ns
GO: BP	GO:0099173	postsynapse organization	150	72	6.64 × 10^−5^	0.136			ns
GO: BP	GO:1902285	semaphorin-plexin signaling pathway involved in neuron projection guidance	11	11	7.18 × 10^−5^	0.136			ns
GO: CC	GO:0071944	cell periphery	5163	1465	9.25 × 10^−5^	0.162			ns
GO: MF	GO:0047372	acylglycerol lipase activity	9			ns	9	3.05 × 10^−5^	0.139
KEGG Analysis—Dichotomized Pb							KEGG—Continuous Pb		
KEGG	path:hsa04360	Axon guidance	174	86	4.76 × 10^−5^	0.016			ns
KEGG	path:hsa04727	GABAergic synapse	84	43	9.06 × 10^−5^	0.153			ns

GO = Gene Ontology; BP = Biological Process; CC = Cellular Component; MF = Molecular Function; KEGG = Kyoto Encyclopedia of Genes and Genomes; Pb = Lead; # = number; ns = not significant (q > 0.2).

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
