# Peer review of "Neonatal Lead (Pb) Exposure and DNA Methylation Profiles in Dried Bloodspots"

_ijerph, 2020, doi:10.3390/ijerph17186775_

Round 1

Reviewer 1 Report

Comments:

Although some of the samples were not collected at the same time, I thought the authors have addressed this limitation well. The authors have applied bloodspots techniques to analyze lead levels and DNA methylation profiles and the results have proved the effectiveness of this technique, which can be used in future studies for other contaminants and their health effects. I will rather recommend this manuscript for publication after the following minor comments are addressed.

Specific comments

  1. The lead levels were determined in the three groups of children, but I can not find any descriptions on the difference of their lead levels. I am curious if there is any age-difference in lead levels among these three groups. I may suggest authors can add these results.
  2. Line 247: there should be a space between were33%.
  3. Line 253: I have not found Figure 1. Should it be Table 1?
  4. Table 1, the full name of “na” and “LOD” should be spelled out in a note of Figure 1.
  5. Line 374: I may suggest combining “increased methylation.” and “decreased methylation” to “deregulation of methylation”.
  6. Line 382: should specify what kind is the human disease.

Author Response

Point by point for reviewer #1

Although some of the samples were not collected at the same time, I thought the authors have addressed this limitation well. The authors have applied bloodspots techniques to analyze lead levels and DNA methylation profiles and the results have proved the effectiveness of this technique, which can be used in future studies for other contaminants and their health effects. I will rather recommend this manuscript for publication after the following minor comments are addressed.

The authors thank the reviewer for these thoughts and reviewing the manuscript.

Specific comments

The lead levels were determined in the three groups of children, but I can not find any descriptions on the difference of their lead levels. I am curious if there is any age-difference in lead levels among these three groups. I may suggest authors can add these results.

Thanks for this suggestion. We have added the lead levels within each group to Supplemental Table S1. The levels were not significantly different by group according to ANOVA (p=0.51).

Line 247: there should be a space between were33%.

Space has been added- see line 265

Line 253: I have not found Figure 1. Should it be Table 1?

This figure is at line 123.

Table 1, the full name of “na” and “LOD” should be spelled out in a note of Figure 1.

The changes have been made on all applicable tables.

Line 374: I may suggest combining “increased methylation.” and “decreased methylation” to “deregulation of methylation”.

This change was made and required a combining of two sentences for consistency– see line 400

Line 382: should specify what kind is the human disease.

Language added to specify – see line 414

Reviewer 2 Report

Reviewers Summary:

The study was conducted to understand a potential link between early life lead exposure and changes in DNA methylation level of the infant. Lead exposure was determined by measuring the level of lead in dried blood spots obtained from babies at time of birth. DNA methylation level were obtained via state of the art human Infinium MethylationEPIC Array technology. Overall, the study was conducted very well and the use of dried blood spots is of high clinical relevance. The analyses done are of good quality and the manuscript is written very well and clearly structured. Please find some minor comment in the following.

General comments:

The main concern of the presented analyses is the problem of different storage conditions of dried blood spots (time, temperature). However, as included in the discussion part (see line 417 and following), experimental validation of reliable and comparable data would highly improve data interpretation and should be considered in future studies. First, a comparison of dried blood spots versus fresh blood samples would be of interest and second, different storing temperatures as well as different storing durations.

Dried blood spots from the babies were obtained via heel pick, but the estimated cell counts were obtained using cord blood. A statement about this is missing and also how cell count was obtained. This should be clarified.

Line 53: It must read: …devoted to understand…..

Line 86-99: I would recommend to place this section into the discussion part of the manuscript.

Line 169: There are missing characters: 70 degree Celsisu; 45micro liter; 30 micro liter

Line 183: please state all quality control measures

Line 188: please delete the word “Project” and use instead R Software for Statistical Computing

Line 190: Have data been logarithmically transformed to approach normal distribution in case of unaccomplished requirement?

Line 194: how was blood cell count estimated?

Line 211: please include missing sources literature sources.

Line 212: Did the authors checked for correlation analyses in the recruitment groups separately but adjusted for cell count?

Line 215: please indicate the abbreviation LOD and LOD/sqrt. I guess it is the same meaning as method detection limit? Please clarify.

Line 229: …as recommended by program developers… Please add the appropriate reference.

Line 229-232: This sentence confuses me. What is meant by “the highest p-value included was 0.014”?

Line 245-246: I recommend writing “BMI of 29.3 ± 8.8 kg/m² standard deviation” instead of (SD) BMI.

Line 248-249: Please specify in what sense the subgroup used for EPIC analyses did not differ from the overall cohort. In addition, table 1 needs some edits: table signature is missing and abbreviations are not stated (SD, na, LOD…etc). Furthermore, line 1 indicates the different groups being presented in the table. However, the term “included in Analysis” is not sufficient. Please indicate clearly that this refers to the subgroup used for EPIC Array analyses. Furthermore, recruitment groups 1, 2, 3 should be explained in the subheading of the table. I recommend deleting the “missing” 2% in the Obesity group from Table 1 and just state below the table that the information from 2 moms regarding BMI is missing. The same for race. As for the cell count estimates - How was this done? Form cord blood or from the dried blood samples? If the latter is true, how was this done?

The same appears to supplementary material – abbreviations are missing and clear descriptions within the subheadings of tables or figures are needed.

Line 258: Please indicate the meaning of abbreviation SVD.

Line 259 and supplementary Figure 1: what defines the principle components of DNA methylation data?

Line 285-287: it is not stated in the manuscript, how DNAseI hypersensitivy sites, TF binding sites and so on were evaluated or which source was used.

Supplementary Table 3 might be included in the main text.

Author Response

Point by point for reviewer #2

The study was conducted to understand a potential link between early life lead exposure and changes in DNA methylation level of the infant. Lead exposure was determined by measuring the level of lead in dried blood spots obtained from babies at time of birth. DNA methylation level were obtained via state of the art human Infinium MethylationEPIC Array technology. Overall, the study was conducted very well and the use of dried blood spots is of high clinical relevance. The analyses done are of good quality and the manuscript is written very well and clearly structured. Please find some minor comment in the following.

General comments:

The main concern of the presented analyses is the problem of different storage conditions of dried blood spots (time, temperature). However, as included in the discussion part (see line 417 and following), experimental validation of reliable and comparable data would highly improve data interpretation and should be considered in future studies. First, a comparison of dried blood spots versus fresh blood samples would be of interest and second, different storing temperatures as well as different storing durations.

Thank you for these suggestions for future studies. We agree that this is a very important line of research as more and more scientists use bloodspots for exposure assessment, epigenetics, and more.

Dried blood spots from the babies were obtained via heel pick, but the estimated cell counts were obtained using cord blood. A statement about this is missing and also how cell count was obtained. This should be clarified.

Blood cell counts are estimated via an algorithm developed for EPIC array data. The algorithm uses data at CpG sites with cell-type specific methylation patterns. These CpG sites were determined by using Infinium array data from sorted cord blood cells. This is a widely used algorithm in studies using the Infinium arrays (there is also version based on adult blood). This is mentioned in the methods section (lines 198-200).  “We estimated cell type proportions from each sample using an algorithm based on Infinium data and a reference dataset from sorted cord blood cells [38,39].” It is true – this method was designed based off of cord blood and not sorted neonatal blood samples. The method estimates the proportions of 7 cord blood cell types – natural killer cells, B cells, monocytes, granulocytes, nucleated red blood cells (nRBCs), CD4+ t-cells, and CD8+ t-cells. These cell types can also be found in newborn blood samples, though the concentration of nRBCs rapidly decline in the first week after birth. In the state of Michigan where these samples come from, bloodspots are collected 1 day after birth. The other option is an algorithm based on adult blood cell types. Given the presence of some nRBCs in the first few days of life, we think the cord blood algorithm is more applicable to bloodspots.    

Line 53: It must read: …devoted to understand…..

Corrected on line 53

Line 86-99: I would recommend to place this section into the discussion part of the manuscript.

Thank you for this thoughtful suggestion. We feel that that this paragraph helps to frame one of the takeaway messages of our paper which is the utility of bloodspots. Having some of this information in the intro helps to prime the reader for the discussion. Therefore, we would respectfully reject this suggestion and have left this section in place.

Line 169: There are missing characters: 70 degree Celsisu; 45micro liter; 30 micro liter

These have been corrected on line 170

Line 183: please state all quality control measures

Starting at line 185, the quality control procedure is now described in more depth. ‘Quality control procedures include examining data for outlier samples, (with low intensity overall and/or by overall distribution of methylation across all probes), removing poorly performing samples (with estimated sex mismatch or with >5% of probes with low detection), and removing poorly performing probes (detection p-value>0.05 in at least 5% of samples.’

Line 188: please delete the word “Project” and use instead R Software for Statistical Computing

This correction was made on line 196

Line 190: Have data been logarithmically transformed to approach normal distribution in case of unaccomplished requirement?

Given the nature of DNA methylation generated from EPIC arrays, there are advantages and arguments for or against transforming the data prior to statistical analysis. Outcome data consist of >750,000 separate CpG sites with ‘beta values’ (proportion methylated from 0 to 1). At each of these sites, beta values across subjects are normally distributed at some CpG sites but not all. Some researchers utilize logit-transformed beta values (called M-values) as the outcomes in their models for epigenome-wide studies. However, using M-values has been shown to inflate significant findings at the tail ends of the original beta distribution (meaning, at CpG sites that have DNA methylation levels very close to 0 or very close to 1). Because of these, many epigenetic epidemiologists prefer to model the original beta values instead. That is what we have done here.

Line 194: how was blood cell count estimated?

Blood cell counts are estimated via an algorithm developed for EPIC array data. The algorithm uses data at CpG sites with very cell-type specific methylation patterns. These CpG sites were determined by using Infinium array data from sorted cord blood cells. This is a widely used algorithm in studies using the Infinium arrays (there is also version based on adult blood). This is mentioned in the methods section (lines 196-198).  “We estimated cell type proportions from each sample using an algorithm based on Infinium data and a reference dataset from sorted cord blood cells [38,39].”

Line 211: please include missing sources literature sources.

We have clarified this part to refer to evidence within the population of this study. “While none of the available covariates had evidence for confounding (associated with both Pb and DNA methylation data), we included sex, race, and estimated cell type proportions (granulocytes, nucleated red blood cells, B cells and monocytes) as covariates in the final models due to their highly significant associations with DNA methylation profiles and evidence for their influence on DNA methylation in this study.”

Line 212: Did the authors checked for correlation analyses in the recruitment groups separately but adjusted for cell count?

We did not test the association separately between DNA methylation and Pb in each recruitment group, adjusted for cell type. We did not do this because the sample size is quite small when stratified by recruitment group. 

Line 215: please indicate the abbreviation LOD and LOD/sqrt. I guess it is the same meaning as method detection limit? Please clarify 

Thank you for point this out – discrepancy between two authors’ contributions. MDL will be consistently used.

Line 229: …as recommended by program developers… Please add the appropriate reference.

Citation has been added (#44) on line 247

Line 229-232: This sentence confuses me. What is meant by “the highest p-value included was 0.014”?

This sentence has been updated for clarity: ‘We also assessed whether any biological pathways were enriched amongst the top 10,000 CpG sites (by raw p-values ranging from 1.33e-6 to 0.014; [44]). For the association with continuous and categorical Pb measures using a gene ontology enrichment algorithm designed for Infinium data that adjusts for the number of CpG sites within each gene represented on the array [44].’ Lines 246-250

Line 245-246: I recommend writing “BMI of 29.3 ± 8.8 kg/m² standard deviation” instead of (SD) BMI.

Corrected on line 264

Line 248-249: Please specify in what sense the subgroup used for EPIC analyses did not differ from the overall cohort. In addition, table 1 needs some edits: table signature is missing and abbreviations are not stated (SD, na, LOD…etc). Furthermore, line 1 indicates the different groups being presented in the table. However, the term “included in Analysis” is not sufficient. Please indicate clearly that this refers to the subgroup used for EPIC Array analyses. Furthermore, recruitment groups 1, 2, 3 should be explained in the subheading of the table. I recommend deleting the “missing” 2% in the Obesity group from Table 1 and just state below the table that the information from 2 moms regarding BMI is missing. The same for race. As for the cell count estimates - How was this done? Form cord blood or from the dried blood samples? If the latter is true, how was this done?

Abbreviations added to table 1 legend and other appropriate tables/figures

“MethylationEPIC” added to table 1 subheading

Group information added to table 1 legend

Deleted “missing” row

For the cell count estimates, refer to explanation of previous comment. In summary: ‘(lines 198-200).  “We estimated cell type proportions from each sample using an algorithm based on Infinium data and a reference dataset from sorted cord blood cells [38,39].” ’

The same appears to supplementary material – abbreviations are missing and clear descriptions within the subheadings of tables or figures are needed.

This has been corrected

Line 258: Please indicate the meaning of abbreviation SVD.

This has been corrected line 276

Line 259 and supplementary Figure 1: what defines the principle components of DNA methylation data?

The Single Value Decomposition method was used to create principle components explaining the largest portions of variability across the entire dataset (all CpG sites). The citation is included in the methods section and is listed here.

Teschendorff, A.E.; Menon, U.; Gentry-Maharaj, A.; Ramus, S.J.; Gayther, S.A.; Apostolidou, S.; Jones, A.; Lechner, M.; Beck, S.; Jacobs, I.J.; et al. An epigenetic signature in peripheral blood predicts active ovarian cancer. PLoS ONE 2009, 4, e8274, doi:10.1371/journal.pone.0008274.

Line 285-287: it is not stated in the manuscript, how DNAseI hypersensitivy sites, TF binding sites and so on were evaluated or which source was used.

Data on genomic features at or near each CpG site comes from the annotation package generated for this EPIC array. The name of the package and information it contains is now listed in the methods section. Lines 192-194

Supplementary Table 3 might be included in the main text.

Supplemental Table 3 is now Table 3

Supplemental Table 4 is now Supplemental Table 3

Table 3 is now Table 4

Language corrected throughout manuscript to match the above changes

Reviewer 3 Report

This type of research showing in utero lead exposure alters the epigenome of the offspring later in life has already been done. However, having said that the novelty of this paper is the detection of lead in different age groups of children including increased methylation, decreased methylation or increased variability with detailed bioinformatics analysis. Due to the small sample size, there can be limitations in playing with the gene expression. Hence, the reviewer feels this manuscript is written and presented to its maximum capacity and does not need any further modifications. 

Author Response

Point by point for reviewer #3

This type of research showing in utero lead exposure alters the epigenome of the offspring later in life has already been done. However, having said that the novelty of this paper is the detection of lead in different age groups of children including increased methylation, decreased methylation or increased variability with detailed bioinformatics analysis. Due to the small sample size, there can be limitations in playing with the gene expression. Hence, the reviewer feels this manuscript is written and presented to its maximum capacity and does not need any further modifications. 

The authors thank the reviewer for their comments and for reviewing the manuscript.